# Emerging Role and Mechanism of the *FTO* Gene in Cardiovascular Diseases

**DOI:** 10.3390/biom13050850

**Published:** 2023-05-17

**Authors:** Zi-Yang Xu, Xia Jing, Xing-Dong Xiong

**Affiliations:** Guangdong Provincial Key Laboratory of Medical Molecular Diagnostics, The First Dongguan Affiliated Hospital, Guangdong Medical University, Dongguan 523808, China; xzy@gdmu.edu.cn (Z.-Y.X.); jingxia@gdmu.edu.cn (X.J.)

**Keywords:** *FTO*, SNP, m6A, cardiovascular diseases

## Abstract

The fat mass and obesity-associated (*FTO*) gene was the first obesity-susceptibility gene identified through a genome-wide association study (GWAS). A growing number of studies have suggested that genetic variants of *FTO* are strongly associated with the risk of cardiovascular diseases, including hypertension and acute coronary syndrome. In addition, FTO was also the first N^6^-methyladenosine (m6A) demethylase, suggesting the reversible nature of m6A modification. m6A is dynamically deposited, removed, and recognized by m6A methylases, demethylases, and m6A binding proteins, respectively. By catalyzing m6A demethylation on mRNA, FTO may participate in various biological processes by modulating RNA function. Recent studies demonstrated that FTO plays a pivotal role in the initiation and progression of cardiovascular diseases such as myocardial fibrosis, heart failure, and atherosclerosis and may hold promise as a potential therapeutic target for treating or preventing a variety of cardiovascular diseases. Here, we review the association between *FTO* genetic variants and cardiovascular disease risk, summarize the role of FTO as an m6A demethylase in cardiovascular disorders, and discuss future research directions and possible clinical implications.

## 1. Introduction

With continued population aging and lifestyle changes, cardiovascular diseases (CVDs) have become an increasing public health concern. Despite significant progress in their diagnosis and treatment, CVDs remain the leading cause of death worldwide [1]. Therefore, increasing our understanding of their pathogenesis, novel diagnostic biomarkers, and effective therapeutic targets is urgently required to reduce the global social and economic impacts of CVDs.

In 2007, the fat mass and obesity–associated (*FTO*) gene was reported as closely associated with body mass index (BMI) and obesity susceptibility [2]. It is located on chromosome 16q12.2, spans 410.50 kb, and contains nine exons. FTO is widely expressed in human tissues such as the adipose tissue, heart, and brain. Further investigation revealed that FTO belongs to the non-heme Fe(II)/α-ketoglutarate-dependent AlkB family of demethylases, which mainly catalyzes the demethylation of 3-methylthymidine in single-stranded DNA and 3-methyluracil in single-stranded RNA [3,4,5]. A breakthrough investigation revealed that FTO efficiently demethylated N^6^-methyladenosine (m6A) in messenger RNA (mRNA), demonstrating the dynamic reversibility of m6A modifications [6]. Since then, the distinct roles of FTO have gradually been revealed. The localization of FTO within the nucleus and cytoplasm varies according to the cell type and affects its ability to perform different RNA modifications. FTO mediates snRNA m6A and cap N6, 2-o-dimethyladenosine (m6Am) demethylation in the nucleus, and mRNA cap m6Am demethylation in the cytoplasm [7]. m6A is the most abundant internal modification in eukaryotic mRNA and the most favorable substrate for FTO [8]. It is regulated by m6A writers, erasers, and readers to control mRNA fate by regulating mRNA splicing, nuclear export, translation, or stability [9,10,11,12,13].

Emerging evidence has indicated that FTO exerts crucial effects on various diseases, including CVDs [14,15,16]. This review focused on the functional role and molecular mechanisms of FTO in CVDs, including myocardial fibrosis, heart failure, and atherosclerosis, providing a new direction for further research on the pathogenesis and treatment of CVDs.

## 2. *FTO* Genetic Variants and CVD Risk

Single-nucleotide polymorphisms (SNPs) are DNA sequence polymorphisms caused by a single-nucleotide substitution mutation at the genomic level, which occur in at least 1% of the population. SNPs are the most common genetic DNA sequence variations, constituting almost 90% of genetic variations in the human genome, with an average of one genotypic polymorphic SNP per thousand bases and an estimated total of up to 3 million [17]. SNPs typically come in two forms: those located within the coding region of genes and those that exist in non-coding regions. SNPs in any region of the gene can potentially affect the protein structure or expression level of the gene product and thus alter an individual’s susceptibility to human disease. Genome-wide association studies (GWAS) is a research method for exploring the association between genes and diseases, using SNPs as genetic markers, aiming to discover genetic variants associated with specific traits and complex diseases at the genetic level. The *FTO* gene is located on human chromosome 16q12.2, including nine exons and eight introns. GWAS have shown that the A allele of the intron *FTO* variant rs9939609 increases the risk of obesity [2]. Because obesity is a significant risk factor for CVDs [18], FTO may play an important role in the development and progression of CVDs. Accumulating evidence has revealed that genetic variants of the *FTO* gene are associated with CVD risk (Table 1).

### 2.1. Incident CVD

Incident CVD was a composite outcome defined as the first occurrence of heart failure, coronary heart disease, or transient ischemic attack. Several studies have investigated the association between *FTO* variants and the risk of incident CVD [19,20,21,22]. Ahmad et al. reported that the *FTO* rs8050136 variant was significantly associated with the risk of incident CVD in Caucasian women, mediated by BMI [19]. However, *FTO* rs8050136 was strongly associated with the risk of incident CVD, even after the adjustment for BMI, in a Danish population [20]. Similarly, several studies have confirmed a significant association between *FTO* rs9939609 and the risk of incident CVD independent of BMI and other conventional CVD risk factors [21,22,23]. These studies suggested that *FTO* variants are likely to affect CVD risk in ways other than BMI. Aijala et al. suggested that one possible mechanism by which *FTO* gene variants affect cardiovascular risk may involve the regulation of FTO expression, thus affecting mTOR signaling that regulates autophagy [23,47].

### 2.2. Hypertension

A growing number of studies have investigated the effects of *FTO* gene variations on the risk of hypertension. Several studies have shown that *FTO* rs9939609 is associated with the risk of hypertension dependent on BMI in studies with different populations [24,25,26,27,29]. This polymorphism may be related to the general mechanism of regulating hypertension, thereby exhibiting susceptibility to hypertension in various populations. Cecil et al. reported that *FTO* is highly expressed in the hypothalamus, which regulates energy homeostasis and metabolism [48]. Thus, the correlation between *FTO* variants and hypertension risk may be related to the regulation of the sympathetic vasomotor tone. There is evidence that *FTO* rs9302652 increases the risk of hypertension, which appears to be related to the higher sympathetic modulation of the vasomotor tone in French Canadians [30]. Gamma-glutamyl transferase could play notable regulatory roles in hypertension-related vascular remodeling. Falbova et al. suggested that the association between *FTO* rs17817449 and hypertension risk may be related to serum gamma-glutamyl transferase levels in middle-aged Slovak women [31]. Another study reported that *FTO* rs8050136, rs9939609 and rs9926289, and *GNB3* rs1129649 and rs5443 were positively associated with essential hypertension in an Indian population [32]. Interactions between genetic variants of *FTO* and *GNB3* influence clinical parameters to augment hypertension, probably by modulating the FTO expression in metabolically relevant tissues such as the hypothalamus and affecting the subsequent translation of key signaling molecules such as *GNB3* [32]. Goulet et al. showed that *FTO* rs9930333 is not significantly associated with systolic blood pressure in Canadian adolescents [33]. These results indicated that different *FTO* genetic variants may influence the risk of hypertension through different mechanisms.

### 2.3. Acute Coronary Syndrome

Acute coronary syndrome (ACS) is caused by atherosclerotic plaque rupture-induced thrombosis or obstruction of the coronary artery, including myocardial infarction and unstable angina. Hubacek et al. found that the *FTO* rs17817449 variant increases the risk of ACS, probably by affecting DNA methylation [34]. Specifically, *FTO* gene variants may interact with unhealthy lifestyle factors (e.g., poor eating habits, smoking, or lack of physical activity), thus affecting epigenetic status and ultimately promoting the development of ACS [34]. Changes in epigenetic features (DNA methylation, histone modifications, and RNA-based mechanisms) have been described as next-level players between SNPs and disease [49,50]. Epigenetic mechanisms in cardiovascular pathophysiology may represent an important association between the genotype and phenotype and may explain some variability. Variants in the first *FTO* intron have been reported to correlate with methylation ability, while differences in methylation status are associated with many morbidities [51,52].

### 2.4. Aortic Valve Stenosis

Aortic valve stenosis (AVS) is a structural heart disease caused by aortic valve hemodynamic obstruction. The aortic valve is located between the left ventricle and the aorta. When the heart contracts, blood flow will flow from the left ventricle into the aorta. Thickening and calcification of the aortic valve will lead to reduced opening and the development of left ventricular outflow obstruction, forcing the heart to pump blood harder. Thron et al. found an association of *FTO* rs9939609 with AVS and suggested that *IRX3*, as a functional long-range target of *FTO* SNPs, may lead to altered CX43 expression, influencing the progression of AVS [35]. Small et al. identified a novel association between rs11647020, a variant in intron 1 of the *FTO* gene, and AVS risk and provided a possible functional mechanism whereby this intronic variant may affect AVS pathobiology independently of BMI through *RBL2*-mediated telomere dysregulation [36].

### 2.5. Others

Venous thromboembolism is a multifactorial disease caused by hereditary and acquired risk factors, including deep venous thrombosis and pulmonary embolism. Two studies reported that *FTO* rs9939609 and rs1558902 are associated with an increased risk of venous thromboembolism [37,38]. Myocardial infarction is defined as myocardial ischemia or necrosis caused by coronary artery occlusion. Studies have reported that *FTO* rs17817449 and rs9939609 play a significant role in the risk of myocardial infarction [39,40,41]. Several studies reported that rs9939609, the most common variant of *FTO*, is associated with the risk of ischemic heart disease and coronary artery disease [42,43,44]. However, two studies showed that *FTO* genetic variants, including rs9939609, rs8050136, and rs9937053, are not associated with stroke risk [45,46]. These findings suggest that *FTO* genetic variants have different susceptibilities to various types of CVDs, but the exact mechanism by which it occurs has yet to be elucidated.

## 3. Role and Mechanism of FTO in CVDs

FTO was not only the first identified obesity risk gene, but it was the first discovered m6A demethylase that catalyzes methyl group removal from the N6 position of adenosine residues [6]. m6A modification is subject to reversible and dynamic regulation by three different types of protein complexes, m6A writers, erasers, and readers, thereby controlling mRNA fate by regulating nuclear export, degradation, stability, and translation [9,10,11,12,13]. An increasing number of studies have indicated that FTO can mediate the m6A demethylation of target mRNAs to affect the initiation and progression of CVDs, such as myocardial fibrosis, heart failure, atherosclerosis, and aortic aneurysm (Figure 1).

### 3.1. Myocardial Fibrosis

Myocardial fibrosis is a major pathological feature of ventricular remodeling that can result in cardiac dysfunction. Recent studies have shown that FTO plays an essential role in the progression of myocardial fibrosis [53,54,55]. Li et al. showed that circCELF1 regulates DKK2 expression via FTO-mediated m6A demethylation and miR-636 to inhibit the progression of myocardial fibrosis [53]. Similarly, Gao et al. reported that FTO-dependent m6A modification regulated cardiomyocyte fibrosis via the PI3K/AKT/GLUT2, PPARγ/RXRα, and mitochondrial apoptosis pathways [54]. Furthermore, another study reported that FTO overexpression in mouse models of myocardial infarction reduced cardiac fibrosis and promoted angiogenesis [55]. These results may provide new options for the treatment of myocardial fibrosis.

### 3.2. Myocarditis

Myocarditis is localized or diffuse inflammation of the myocardial tissue caused by different insults, such as infections, autoimmune reactions, toxins, and adverse drug reactions. Yu et al. demonstrated that FTO regulates m6A modification and expression levels of cardiac proinflammatory cytokines to attenuate myocardial inflammation and dysfunction during endotoxemia [56]. However, Dubey et al. showed that the loss of function of FTO impaired the stability of CD36 mRNA and inhibited fatty acid–induced cardiomyocyte inflammation [57]. These observations suggested that FTO expression and function differ under different pathological conditions.

### 3.3. Ischemia–Reperfusion Injury

Ischemia–reperfusion (I/R) injury refers to exacerbated and irreversible tissue damage caused by blood flow restoration after a prolonged period of occlusion in clinical conditions, such as myocardial infarction, stroke, and post-transplantation. I/R activates multiple cell death pathways, including necrosis, apoptosis, or autophagy-associated cell death. The inhibition of apoptosis may be a promising therapeutic strategy for I/R injury. Ke et al. demonstrated that the m6A demethylase FTO attenuated cardiomyocyte apoptosis and inflammation by increasing YAP1 mRNA stability in myocardial I/R injury [58]. Xu et al. reported that FTO demethylates Bcl2 transcript and enhances Bcl2 expression, which alleviates neuronal apoptosis in cerebral I/R injury [59]. Oxidative stress is an oxidative and antioxidant imbalance mediated by reactive oxygen species, leading to oxidative damage to tissues and cells, which plays a major role in I/R injury. A recent study reported that FTO enhances Nrf2 expression by mediating the m6A demethylation of Nrf2 mRNA, thus inhibiting the oxidative stress response and ultimately alleviating cerebral I/R injury [60]. Du et al. showed that the m6A demethylase FTO ameliorates hepatic I/R injury by alleviating liver oxidative stress and Drp1-mediated mitochondrial fragmentation [61]. In addition, FTO reduced damage to and promoted the recovery of ischemic tissue. It was recently reported that the m6A demethylase FTO decreases poststroke gray and white matter damage and improves motor function recovery [62]. Li et al. confirmed that FTO catalyzes the demethylation of phospholipid phosphatase 3 mRNA, thereby increasing its expression and enhancing vascular repair after ischemic stroke [63]. Collectively, these results suggested that FTO alleviates I/R injury by reducing apoptosis, inhibiting oxidative stress responses, and promoting ischemic tissue repair.

### 3.4. Heart Failure

Heart failure is characterized by abnormal left ventricular filling or ejection caused by the impairment of cardiac structure and function. Berulava et al. reported that the cardiomyocyte-specific knockout of the demethylase FTO led to the faster progression of heart failure, a significantly reduced ejection fraction, and increased dilatation [64]. Zhang et al. reported that the m6A demethylase FTO was significantly upregulated in the peripheral blood samples of patients with heart failure and might serve as a novel diagnostic biomarker of heart failure [65]. Cardiomyocyte loss and the lack of contractility are the major pathogenic mechanisms that lead to heart failure. Therefore, the inhibition of apoptosis to reduce cardiomyocyte loss has potential implications in the treatment of heart failure. Shen et al. demonstrated that FTO overexpression inhibited the apoptosis of hypoxia–reoxygenation–treated myocardial cells by regulating the m6A modification of Mhrt, thereby improving heart failure [66]. Recent studies have shown that FTO effectively enhances the contractile function of cardiomyocytes, thereby regulating heart failure. Mathiyalagan et al. discovered that FTO is downregulated in failed mammalian hearts and hypoxic cardiomyocytes, whereas the overexpression of FTO selectively demethylates cardiac contractile transcripts, thus restoring their protein expression and increasing the contractile function of cardiomyocytes [55]. Consistent with this, a recent study reported that FTO downregulation reduced the contraction of hypoxic cardiomyocytes, resulting in heart failure [67]. Another study showed that the m6A demethylase FTO attenuated cardiomyocyte contractile dysfunction by regulating glucose uptake and glycolysis in mice with pressure overload-induced heart failure [68]. These findings confirmed that FTO can delay the progression of heart failure and provide new avenues for the development of diagnostic and therapeutic strategies for heart failure.

### 3.5. Atherosclerosis

Atherosclerosis is a chronic inflammatory vascular disease characterized by the progressive accumulation of lipids, inflammatory cells, and fibrous materials in the arterial wall that can narrow blood vessels. Activated macrophages take up lipids and transform them into macrophage-derived foam cells that play key roles in the initiation and progression of atherosclerosis. A recent study showed that FTO regulates cholesterol deposition in macrophage foam cells, thereby preventing atherosclerotic plaque formation [69]. On the one hand, FTO-dependent m6A demethylation downregulates CD36 expression by suppressing PPARγ, resulting in the inhibition of macrophage lipid uptake. On the other hand, FTO increases ABCA1 expression in an AMPK activity-dependent manner, which accelerates intracellular cholesterol efflux [69]. Another study revealed that FTO regulated the expression of atheroprotective genes (eNOS and KLF2) in an m6A-dependent and YTHDF3-mediated manner, thus affecting endothelial cell inflammatory responses [70]. These observations suggest that FTO regulates various pathways involved in atherosclerotic progression by mediating target RNA demethylation.

### 3.6. Hypertension

Hypertension is a common chronic disorder that increases the risk of several cardiovascular disorders, including stroke, heart failure, and coronary heart disease. Kruger et al. showed that the loss of endothelial FTO preserved the myogenic tone in resistant arteries by increasing PGD2 levels, contributing to protection from obesity-induced hypertension [71]. This newly discovered mechanism provides clear evidence of the involvement of FTO in the regulation of hypertension.

### 3.7. Abdominal Aortic Aneurysm

Abdominal aortic aneurysm (AAA) is a degenerative vascular disease characterized by the permanent dilatation of the aorta with structural malformations within all layers of the vascular wall. AAA formation mainly involves extracellular matrix degradation, smooth muscle activation, and inflammatory cell infiltration. A recent study found a strong correlation between FTO and aneurysmal smooth muscle cell and macrophage infiltration, indicating that FTO may play an important role in AAA progression [72]. Hence, a better understanding of the molecular mechanisms by which FTO regulates smooth muscle activation and the infiltration of inflammatory cells could aid in the discovery of novel therapeutic approaches for aneurysms.

### 3.8. Arrhythmia

Arrhythmia is a disease in which disturbances in the frequency or rhythm of heart excitement are caused by abnormal cardiac electrophysiological activity generated by the electrical conduction system of the heart. In a recent study, the global knockout of FTO led to an imbalance in the autonomic neural modulation of cardiac function in the sympathetic direction and potentially proarrhythmic remodeling of the myocardium [73]. This finding provides new insight into myocardial protection and autonomic neural regulation and sheds new light on potential therapeutic targets for arrhythmia.

The functional role of FTO in CVDs is summarized in Table 2.

## 4. Conclusions and Perspectives

*FTO* was originally identified as an obesity susceptibility gene in genome-wide association studies [2]. Subsequently, correlations between genetic variants of *FTO* and CVDs, such as hypertension, myocardial infarction, and ACS, have also been reported [24,34,41]. Therefore, in-depth research on FTO will aid the prevention and treatment of CVDs. Further studies discovered that FTO is an m6A demethylase that regulates m6A RNA modification, which is involved in many basic physiological processes [6]. More importantly, FTO-dependent m6A modification plays a key role in the initiation and progression of various CVDs and may provide new options for early diagnosis and treatment [54,57,58].

Although the correlation between *FTO* polymorphism and CVD risk is definite, the specific mechanisms of these SNPs in CVD pathogenesis have been elusive. A few studies have investigated the mechanism of action of *FTO* SNPs in obesity. For example, Claussnitzer et al. reported that *FTO* rs1421085 can disrupt ARID5B repressor binding, leading to the derepression of *IRX3* and *IRX5* during early adipocyte differentiation [74]. *IRX3* and *IRX5* have also been found to regulate cardiac impulse conduction, thereby affecting arrhythmias, cardiac remodeling, and reduced cardiac function [75,76]. Stratigopoulos et al. found that *FTO* rs8050136 could regulate the expression of *FTO* and *RPGRIP1L* in the brain by binding to the transcription factor CUX1, thereby influencing leptin signaling, leading to obesity [77]. Based on these findings, *FTO* SNPs may influence CVD risk by regulating the expression of *FTO* or adjacent genes. However, more convincing and systematic research studies are needed to decipher the causal mechanism between *FTO* variants and CVDs.

FTO plays a critical role in the pathogenesis of various diseases. Therefore, the development of FTO modulators for potential therapeutic applications is vital. The elucidation of the FTO crystal structure revealed the mechanisms of substrate recognition and catalytic demethylation, which facilitated the development of small-molecule inhibitors targeting FTO. To date, more than a dozen FTO inhibitors with varying potencies and specificities have been developed [78]. As a member of the non-heme dioxygenase superfamily, FTO catalyzes Fe(II)- and 2-oxoglutarate (2-OG)-dependent oxidative substrate demethylation [3]. Thus, according to their mechanism of action, FTO inhibitors can be divided into three main categories: iron ion chelators, 2-OG-competitive inhibitors, and substrate-competitive inhibitors [79]. Most FTO inhibitors are substrate-competitive inhibitors that selectively and effectively inhibit FTO m6A demethylase activity. Some inhibitors affect various diseases, including CVDs, by targeting FTO [56,71,80]. Kruger et al. found that incubating arteries from obese individuals with the FTO inhibitors rhein or FB23-2 could increase the myogenic tone and protect against obesity-induced hypertension [71]. Yu et al. demonstrated that the chemically modified monomeric compound LHD was an FTO inhibitor that could bind to FTO and interfere with FTO-mediated m6A modification, thereby regulating inflammatory responses in cardiomyocytes [56]. Disease treatment should maximize the therapeutic effect while minimizing side effects. Several studies have preliminarily evaluated the potential side effects of FTO inhibitors, indicating that these drugs can inhibit tumor cell activity and have little effect on healthy cells over a range of concentrations [80,81,82]. However, systematic in vivo toxicity studies are required to assess the potential long-term side effects of small-molecule inhibitors targeting FTO. In the future, additional FTO inhibitors should be validated in clinical trials to provide safer and more effective treatment options. Given that FTO upregulation can delay the progression of several diseases, including myocardial fibrosis, heart failure, and stroke, the development of FTO activators is also necessary. These FTO modulators will provide new strategies for intervening with FTO activity and will likely make FTO a promising therapeutic target.

As studies on FTO have progressed, controversy has increased regarding the use of FTO substrates. Mauer et al. suggested that FTO preferentially demethylates m6Am rather than m6A [83]. To date, little is known about the role of m6Am in determining RNA fate. Although a previous study claimed that m6Am-modified RNAs are more stable, FTO knockdown did not significantly affect the transcript levels of these RNAs [7]. Moreover, the abundance of m6A was much higher than that of m6Am, and the change in m6A was more obvious in FTO-knockdown cell lines, indicating that m6A is the main substrate of FTO [7,8,84]. Further independent studies have shown that m6Am modification promotes or inhibits the translation of target mRNA [85,86,87]. The different effects of m6Am on target RNA stability or translation may be associated with the binding of different m6Am readers. In addition to modifying m6A and m6Am, FTO can also bind to tRNA, act as an m1A demethylase of tRNA, and affect protein translation efficiency [7]. Given that tRNA is a very abundant RNA molecule in cells and that m1A modification is important for cell viability and fitness, it is necessary to investigate the function of FTO as an m1A demethylase in the future.

## Figures and Tables

**Figure 1 biomolecules-13-00850-f001:**
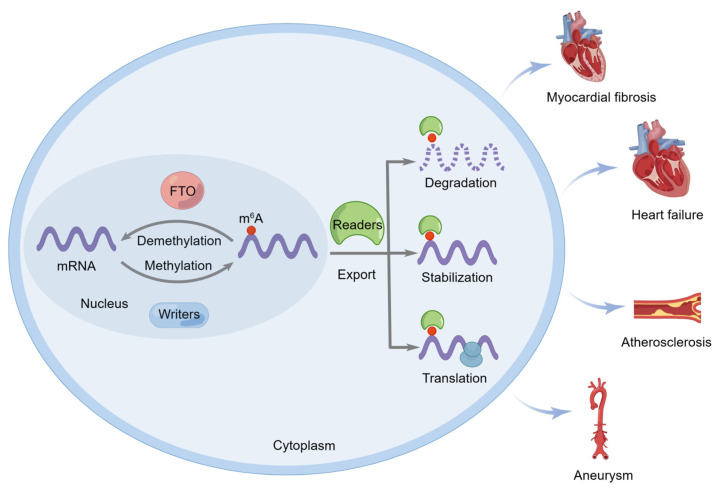
The fat mass and obesity–associated (FTO) protein regulates the initiation and progression of cardiovascular diseases. FTO can catalyze the m6A demethylation of target mRNA and affect its nuclear export, degradation, stability, and translation, thus affecting the initiation and progression of cardiovascular diseases such as myocardial fibrosis, heart failure, atherosclerosis, and aneurysm.

**Table 1 biomolecules-13-00850-t001:** The association between *FTO* genetic variants and CVD risk.

Diseases	SNP	Population	Association	References
Incident CVD	rs8050136	Caucasian, Danish	+	[19,20]
	rs9939609	American, Finnish and Finland	+	[21,22,23]
Hypertension	rs9939609	Danish, Japanese, Chinese, Mexican, Indian	+	[24,25,26,27,28,29]
	rs9302652	Canadian	+	[30]
	rs17817449	Slovak	+	[31]
	rs8050136, rs9926289	Indian	+	[32]
	rs9930333	Canadian	-	[33]
Acute coronary syndrome	rs17817449	Czech	+	[34]
Aortic valve stenosis	rs9939609	German	+	[35]
	rs11647020	White	+	[36]
Venous thromboembolism	rs9939609	Danish	+	[37]
	rs1558902	French, Dutch, American	+	[38]
Myocardial infarction	rs17817449	Czech	+	[39]
	rs9939609	Britisher	+	[40,41]
Ischemic heart disease	rs9939609	Danish	+	[42]
Coronary artery disease	rs9939609	Pakistani	+	[43,44]
Stroke	rs8050136, rs9939609	Chinese	-	[45]
	rs9937053	German	-	[46]

+: indicates significantly related. -: indicates not significantly related.

**Table 2 biomolecules-13-00850-t002:** Functional role of FTO in cardiovascular diseases.

Diseases	Target RNA	Function	References
Myocardial fibrosis	DDK2	Regulated the activation, viability, and migration of cardiac fibroblasts	[53]
	-	Suppressed cardiomyocytes apoptosis and myocardial dysfunction	[54]
Myocarditis	IL-6, IL-1β, TNF-α	Influenced the expression of proinflammatory factors to regulate myocarditis and dysfunction during endotoxemia	[56]
	CD36	Increased CD36 expression and suppressed anti-inflammatory effects	[57]
Ischemia– reperfusion injury	YAP1	Regulated the apoptosis and inflammation of H/R-induced cardiomyocytes	[58]
	Bcl2	Regulated I/R-induced neuronal apoptosis	[59]
	Nrf2	Inhibited the oxidative stress response and ultimately alleviated cerebral I/R injury	[60]
	Drp1	Contributed to the hepatic protective effect via impairing Drp1-mediated mitochondrial fragmentation	[61]
	-	Decreased poststroke gray and white matter damage and improved motor function recovery	[62]
	Plpp3	Contributed to vessel regeneration and constituted a more mature vessel network after stroke	[63]
Heart failure	-	Promoted heart recovery and alleviated HF	[64]
	-	Upregulated in the peripheral blood samples of patients with heart failure and might serve as a novel diagnostic biomarker of heart failure	[65]
	Mhrt	Inhibited apoptosis of H/R-treated myocardial cells	[66]
	SERCA2A, MYH6, RYR2	Restored cardiac contractile protein expression to increase the contractile function of cardiomyocytes, reduce cardiac fibrosis, and boost angiogenesis in ischemic myocardium	[55]
	-	Affected the contractile ability of hypoxic cardiomyocytes	[67]
	Pgam2, Akt-GLUT4	Attenuated cardiomyocyte contractile dysfunction by regulating glucose uptake and glycolysis	[68]
Atherosclerosis	PPARγ	Inhibited macrophage lipid uptake and accelerated intracellular cholesterol efflux	[69]
	eNOS, KLF2	Regulated the expression of atheroprotective genes and affected endothelial cell inflammatory responses	[70]
Hypertension	PDG2	Affected the myogenic tone in resistance arteries	[71]
Aneurysm	-	Associated with aneurysmal smooth muscle cells and might play an important role in the progression of aneurysm	[72]
Arrhythmia	-	Affected the autonomic neural modulation of cardiac function in the sympathetic direction	[73]

FTO, fat mass and obesity-associated; HF, heart failure; H/R, hypoxia–reperfusion; IL, interleukin; I/R, ischemia–reperfusion; MI, myocardial infarction; PPAR, peroxisome proliferator-activated receptor; TNF, tumor necrosis factor.

## Data Availability

Not applicable.

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
