# Peer review of "Emerging Role and Mechanism of the FTO Gene in Cardiovascular Diseases"

_biomolecules, 2023, doi:10.3390/biom13050850_

Round 1

Reviewer 1 Report

This work is very interesting, it is a great job when you try to identify SNPs of genes in CVD framework. In this way, my suggestions are related with the difference in the results by population, although some results looks like same.

Also, for me, it is very curious that some of these results would be negative or positive associations, because all associations are in the frame of CVD.

I think that you can improve your review with more references to justify these associations and try to explain the reasons to obtain same results in studies with different population.

Author Response

Response to Reviewer 1 Comments

Point 1: This work is very interesting, it is a great job when you try to identify SNPs of genes in CVD framework. In this way, my suggestions are related with the difference in the results by population, although some results looks like same. Also, for me, it is very curious that some of these results would be negative or positive associations, because all associations are in the frame of CVD. I think that you can improve your review with more references to justify these associations and try to explain the reasons to obtain same results in studies with different population.

Response 1: Thanks for the constructive suggestion. In the second paragraph of page 3, several studies have shown that FTO rs9939609 is associated with the risk of hypertension dependent on BMI in studies with different populations [24-27,29]. This polymorphism may be related to the general mechanism of regulating hypertension, thereby exhibiting susceptibility to hypertension in various populations (lines 88 to 91 of page 3).

Reviewer 2 Report

This paper raises an interesting issue. However, it is an opinion piece in which the phrase "needs further research" is often repeated. The purpose of an opinion paper is to present the issue in a multidimensional and complete way. This paper lacks an explanation of the mechanism by which FTO polymorphism works. There is a lack of explanation of the signaling pathways that lead to clinically feared changes. This information should be included in the text and especially in the diagrams. Information on conjugations with other genes is also missing.

Author Response

Response to Reviewer 2 Comments

Point 1: This paper raises an interesting issue. However, it is an opinion piece in which the phrase "needs further research" is often repeated. The purpose of an opinion paper is to present the issue in a multidimensional and complete way.

Response 1: As suggested by the reviewer, we have revised “m6Am function requires further investigation.” as “Further independent studies have shown that m6Am modification promotes or inhibits the translation of target mRNA [85-87]. The different effects of m6Am on target RNA stability or translation may be associated with the binding of different m6Am readers.” (lines 354 to 357 of page 10), and revised "the function of FTO as an m1A demethylase deserves further exploration.” as “it is necessary to investigate the function of FTO as an m1A demethylase in the future.” (lines 360 to 361 of page 10).

Point 2: This paper lacks an explanation of the mechanism by which FTO polymorphism works. There is a lack of explanation of the signaling pathways that lead to clinically feared changes. This information should be included in the text and especially in the diagrams. Information on conjugations with other genes is also missing.

Response 2: Thank the reviewer very much for the thoughtful suggestion. Although the correlation between FTO polymorphism and CVD risk is definite, the specific mechanisms of these SNPs in CVD pathogenesis have been elusive. According to these references’ discussion, a few insights about the mechanisms by which FTO SNPs may influence CVD risk have been added to the revised manuscript:

  • Aijala et al. suggested that one possible mechanism by which FTO gene variants affect cardiovascular risk may involve the regulation of FTO expression, thus affecting mTOR signaling that regulates autophagy [23,47] (lines 83 to 85 of page 3).
  • Another study reported that FTO rs8050136, rs9939609 and rs9926289, and GNB3 rs1129649 and rs5443 were positively associated with essential hypertension in an Indian population [32]. Interactions between genetic variants of FTO and GNB3 influence clinical parameters to augment hypertension, probably by modulating the FTO expression in metabolically relevant tissues such as hypothalamus and affecting the subsequent translation of key signaling molecules like GNB3 [32] (lines 100 to 105 of page 3).
  • Thron et al. found an association of FTO rs9939609 with AVS and suggested that IRX3 as a functional long-range target of FTO SNPs may lead to altered CX43 expression, influencing the progression of AVS [35]. Small et al. identified a novel association between rs11647020, a variant in intron 1 of the FTO gene, with AVS risk, and provided a possible functional mechanism whereby this intronic variant may affect AVS pathobiology independently of BMI through RBL2-mediated telomere dysregulation [36] (lines 129 to 134 of page 4).

Reviewer 3 Report

An interesting review about the Role and Mechanism of FTO Gene in Cardiovascular Diseases. This gene and its genetic variants seem to be involved in many cardiovascular issues. Generally, I have not significant comments about the form of the review. It is well-written and has covered the topic from various aspects. I would like to make some minor comments, mainly for some titles: 1) I propose that the authors rephrase the main title of the review, concerning the term "mechanism". They could say implication or something similar. 2) Moreover, in the two Sections entitled as "Others", I believe that they should also rephrase the titles and describe it more analytically.

minor editing

Author Response

Response to Reviewer 3 Comments

Point 1: An interesting review about the Role and Mechanism of FTO Gene in Cardiovascular Diseases. This gene and its genetic variants seem to be involved in many cardiovascular issues. Generally, I have not significant comments about the form of the review. It is well-written and has covered the topic from various aspects. I would like to make some minor comments, mainly for some titles: 1) I propose that the authors rephrase the main title of the review, concerning the term "mechanism". They could say implication or something similar.

Response 1: Thanks for the reviewer’s suggestion. This review mainly focuses on the function and mechanism of FTO as a demethylase, we think that the title “mechanism” is reasonable.

Point 2: Moreover, in the two Sections entitled as "Others", I believe that they should also rephrase the titles and describe it more analytically.

Response 2: Thanks for the suggestion. We have revised “3.6. Others” as three subtitles “3.6. Hypertension ”, “3.7. Abdominal aortic aneurysm” and “3.8. Arrhythmia” (page 7).

Round 2

Reviewer 2 Report

no answer